# A long-range *cis*-regulatory element for class I odorant receptor genes

Tetsuo Iwata[1], Yoshihito Niimura[2,3], Chizuru Kobayashi[4], Daichi Shirakawa[4], Hikoyu Suzuki[5], Takayuki Enomoto[1], Kazushige Touhara[2,3], Yoshihiro Yoshihara[3,6] & Junji Hirota[1,4]

Individual olfactory sensory neurons express a single odorant receptor gene from either class I genes residing in a single cluster on a single chromosome or class II genes spread over multiple clusters on multiple chromosomes. Here, we identify an enhancer element for mouse class I genes, the J element, that is conserved through mammalian species from the platypus to humans. The J element regulates most class I genes expression by exerting an effect over ~ 3 megabases within the whole cluster. Deletion of the *trans* J element increases the expression frequencies of class I genes from the intact J allele, indicating that the allelic exclusion of class I genes depends on the activity of the J element. Our data reveal a long-range *cis*-regulatory element that governs the singular class I gene expression and has been phylogenetically preserved to retain a single cluster organization of class I genes in mammals.

[1] Center for Biological Resources and Informatics, Tokyo Institute of Technology, Yokohama 226-8501, Japan. [2] Department of Applied Biological Chemistry, Graduate School of Agricultural and Life Sciences, The University of Tokyo, Tokyo 113-8657, Japan. [3] ERATO Touhara Chemosensory Signal Project, The Japan Science and Technology Agency, The University of Tokyo, Tokyo 113-8657, Japan. [4] Department of Life Science and Technology, Graduate School of Life Science and Technology, Tokyo Institute of Technology, Yokohama 226-8501, Japan. [5] Nihon BioData Corporation, 3-2-1 Sakado, Takatsu-ku, Kawasaki 213-0012, Japan. [6] RIKEN Brain Science Institute, Saitama 351-0198, Japan. Correspondence and requests for materials should be addressed to J.H. (email: jhirota@bio.titech.ac.jp)

Olfactory sensory neurons (OSNs) in the main olfactory epithelium (MOE) initiate the sense of smell by detecting a vast number of odorous chemicals through odorant receptors (ORs). ORs are G protein-coupled receptors with a putative seven-transmembrane domain structure[1]. OR genes form the largest gene family present in any genome analyzed to date. Phylogenetic analyses separate mammalian OR sequences into two distinct groups, which are referred to as class I and class II ORs[2]. Class I OR genes, which were first identified in fish and then in frog[3, 4], were initially considered to be evolutionary relics in mammals. However, genomic analyses have revealed that mammals possess a substantial number of functional class I OR genes[5], suggesting that class I ORs have physiologically important functions. On the other hand, class II OR genes are specific to terrestrial animals and account for ~ 90% of the mammalian OR repertoire.

Individual OSNs are believed to express a single functional allele of a single OR gene from the repertoire of either class I or class II ORs. The process of the singular OR gene expression occurs through two sequential steps, a single OR gene choice that initiates the process and subsequent maintenance of transcription of the chosen single OR allele. Regarding the initial choice of a single OR gene, Magklara et al. reported that all OR genes are epigenetically silenced prior to OR choice and that a singular OR allele is chosen by stochastic escape from heterochromatic silencing[6]. The latter step, referred to as the "one neuron-one receptor" rule, is maintained by an OR-elicited negative-feedback signal that prevents the activation of additional OR genes[7]. It has recently been demonstrated that this signal is triggered by components of the unfolded protein response, which detect the newly synthesized OR within the endoplasmic reticulum and thereby activate downstream signaling cascades to stabilize OR transcription and to prevent secondary OR gene choice[8]. Although these studies describe a molecular mechanism for ensuring singular OR gene transcription, the cis-regulatory elements/enhancers, which may open up the local chromatin architecture to orchestrate the first step OR choice are poorly understood.

The function of cis-regulatory elements for the transcriptional activation of a single gene within a multigene cluster has been well-documented for the β-globin cluster, in which the locus control region (LCR) located 6–22 kilobases (kb) upstream of the proximal globin gene enhances the transcription of one of five linked β-globin genes depending on the developmental stage of the organism[9]. A few cis-regulatory elements have been reported in mouse class II OR genes. The 2.1-kb H element resides 75 kb upstream of the nearest OR gene, Olfr1507, and functions as the LCR of a cluster of seven class II OR genes spread over 141 kb on chromosome 14[7]. A targeted deletion of the H element abolished the expression of three proximal OR genes and decreased the expression of four distal OR genes in the cluster but did not affect the expression of ORs outside the cluster[10, 11]. The P element regulates the expression of 10 OR genes within the P-centromeric cluster, which consists of 24 OR genes spread over 471 kb on chromosome 7[12].

Because ~ 1100 mouse OR genes are scattered over 69 loci (including singletons) within the genome[13], additional regulatory elements are expected. Recently, a genome-wide search for intergenic OR enhancers by DNase I hypersensitivity-sequencing and chromatin immunoprecipitation sequencing experiments uncovered 35 candidate elements linked to OR clusters, including H and P in mice[14]. The targeted deletion of one of these, Lipsi, led to reduced expression of the eight class II genes within the linked cluster on chromosome 2, whereas OR genes from a distal cluster on the same or different chromosomes were unaffected[14]. Thus, all three regulatory elements for class II genes operate

locally and in cis, controlling 7–10 genes within the adjacent clusters. The distances over which these elements exert effects are ~ 200 kb. To date, no cis-regulatory elements for class I genes have been identified.

During evolution, class I OR genes were maintained in a single cluster and have come to form one of the largest gene clusters in the mouse genome[13]. By sharp contrast, class II OR genes were spread over most chromosomes by duplication and intra- and inter-chromosomal translocation[15]. The differential genomic organization of class I and class II OR genes is enigmatic, and the reason why class I genes did not migrate outside the cluster or to other chromosomes is unknown. In the mouse, 158 class I genes reside in a single huge cluster spread over a ~ 3 megabase (Mb) genomic region on chromosome 7, and this cluster is interrupted by the β-globin cluster[16]. By focusing on one of the mouse class I genes, Olfr544, we previously provided experimental evidence for a cis-regulatory element using Bacillus subtilis genome (BGM) transgenesis[17].

In this study, we analyzed a cis-regulatory element of the mouse class I OR cluster and identified the J element, which exhibits unique features with respect to its long-range regulation and is conserved in all mammalian species examined. The J element regulates class I genes expression of a larger number of genes and over a greater genomic distance than known class II enhancers. Genome analysis uncovers evolutionarily conserved sequence motifs in the J element from the platypus to humans. In addition, we provide genetic evidence that the allelic exclusion of class I genes depends on the activity of the J element. Thus, our data reveal a long-range cis-regulatory element that governs the singular expression of class I gene and has been phylogenetically preserved to retain a single cluster organization of class I genes in mammals.

## Results

**The class I OR cluster.** One of the characteristic features of the class I OR family in mammals is that all class I OR genes reside on a single chromosome, comprising a single huge gene cluster that does not contain any class II genes. In mice, 158 class I genes, including 29 pseudogenes, reside at positions 102,476,773–105,369,317 on chromosome 7 (mm10), forming a ~ 3 Mb cluster (Fig. 1a). The genomic size of the class I OR cluster is ~ 70 times larger than that of the β-globin cluster embedded in this region. There are two phylogenetic groups of class I ORs, the α- and β-groups[18]. The α-group is specific to tetrapods, whereas the β-group is conserved across vertebrate species[19]. In mice, the β-class I OR family consists of three genes, Olfr543, Olfr544, and Olfr545, which are located near the centromeric end of the class I cluster (Fig. 1a, b). These OR genes have intact open reading frames and belong to the MOR42 family.

**A cis-regulatory element for the class I OR gene Olfr544.** Because β-class I ORs are phylogenetically most well-conserved among vertebrates, we focused on this group to examine whether a cis-regulatory element is located in proximity to the β-class I genes. Using BGM transgenesis, we recently reported the presence of a cis-regulatory element for the Olfr544 β-class I gene[17]. Conservation analysis of the genomic region of the transgene using VISTA revealed conserved synteny of the coding regions of class I OR genes and neighboring genes (Stim1, Rrm1, and Trim21) among placental mammals including human, macaque, horse, dog, and cow (Fig. 1b). In addition, we found a strikingly homologous intergenic region between α- and β-class I genes in all analyzed mammalian genomes but not in zebrafish or Xenopus tropicalis (Fig. 1b, Supplementary Fig. 1a).

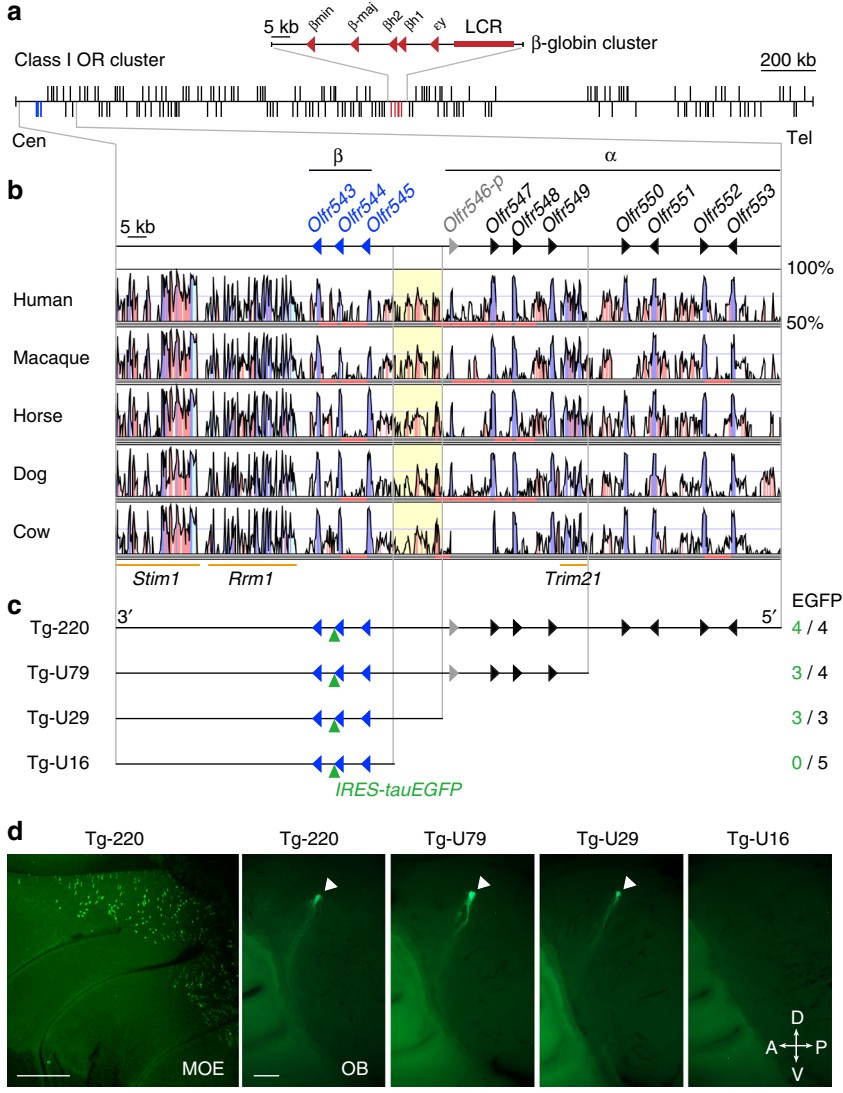

**Fig. 1** Genomic organization of the class I OR cluster and identification of a *cis*-regulatory element. **a** Genomic region of the mouse class I OR gene cluster on chromosome 7. The *vertical lines* represent the positions of 158 class I OR genes (*black*) and 5 β-globin genes. Genes transcribed in the centromeric (Cen) to the telomeric (Tel) direction are drawn above the *horizontal line*; those transcribed in the telomeric to centromeric direction are drawn *below*. The *red arrowheads* depict the transcriptional orientation of β-globin genes, and a *red box* represents the LCR of the β-globin cluster. **b** Nucleotide percent identity plots of the centromeric end of the class I OR gene clusters of mouse and other mammals using VISTA. The *arrowheads* indicate the transcriptional orientations of β-class I OR genes (*blue*) and α-class I OR genes (intact genes, *black*; a pseudogene, *gray*). Highly conserved regions ( > 70% identity) in coding exons, noncoding exons and intergenic regions are shown in *blue*, *cyan*, and *pink*, respectively. The *orange bars* indicate the non-OR genes *Trim21*, *Rrm1*, and *Stim1*. **c** Deletion series of *Olfr544* transgenes. The number of EGFP-positive independent founders and lines among the total analyzed is shown on the right. **d** Representative whole-mount images of the medial aspect of the MOE (Tg-220) and the OB of Tg mice from the deletion series. *Arrowheads* indicate each glomerulus. The *scale bar* is 500 μm

To more precisely identify the region responsible for *Olfr544* transgene expression, we generated a 5′ deletion series based on a Tg-220 transgene in which *Olfr544* was tagged with an *IRES-tauEGFP* reporter (Fig. 1c). Referring to the VISTA plots, we truncated the 5′ end to yield shorter transgenes with 79 kb (Tg-U79), 29 kb (Tg-U29), and 16 kb (Tg-U16) of genomic DNA upstream of the transcription start site (TSS) of *Olfr544*. All four lines carrying Tg-220 showed a punctate pattern of enhanced green fluorescent protein (EGFP) expression within the dorsal zone of the MOE and axonal projections into the dorsal glomeruli of each olfactory bulb (OB) (Fig. 1d). In a deletion series, Tg-U79 and Tg-U29 also showed robust transgene expression with consistent glomerular convergence, whereas none of the Tg-U16 lines or founders showed EGFP expression. These results indicate that a 13-kb intergenic region located 16 kb to 29 kb upstream of

the TSS of *Olfr544* contains a *cis*-regulatory element for *Olfr544* expression.

**Identification of a potential enhancer element**. Because critical intergenic regions such as transcriptional enhancers are frequently conserved during evolution, we first searched for homologous sequences between mouse and human genomes to identify a potential enhancer element in the 13-kb region. A VISTA plot of the mouse 13-kb enhancer region to the human genome delineated several homology peaks (Fig. 2a). Next, to examine the conservation of these homology peaks in other mammalian species, we selected one species as a representative of each placental mammalian order: human (Primates), guinea pig (Rodentia), rabbit (Lagomorpha), horse (Perissodactyla), dog

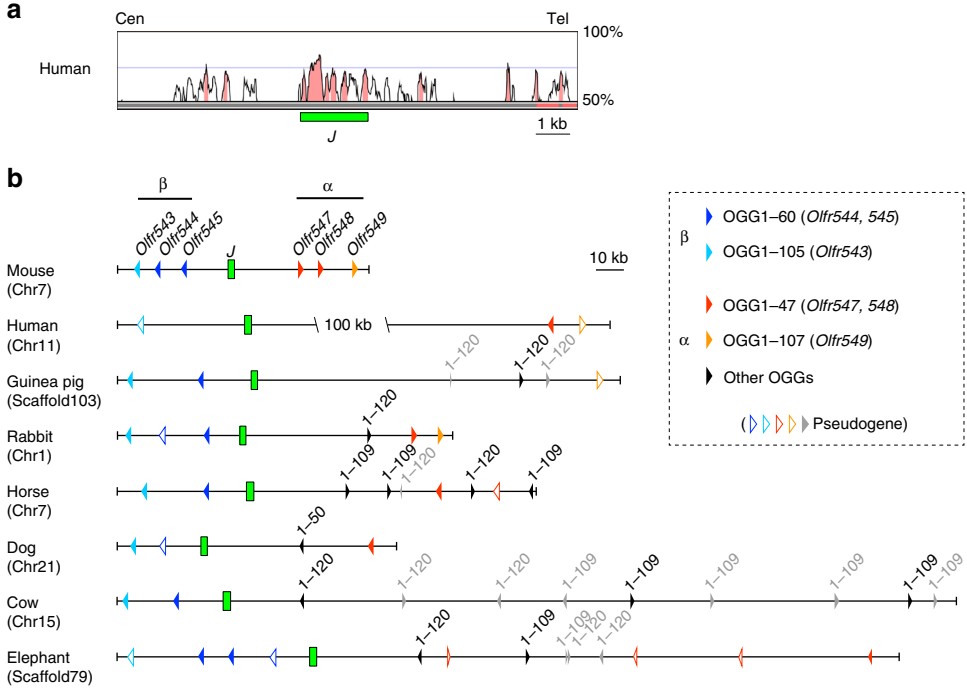

**Fig. 2** The J element is the most highly conserved element among OR enhancers. **a** Nucleotide percent identity plot (VISTA plot) of the junctional region of α- and β-class I OR genes in mouse and human. A 1.9-kb intergenic region, named the J element, was identified as a highly conserved region. **b** Genomic organization of α-, β-class I OR genes and of the J element in placental mammalian species as a representative of each order. Class I OR genes belonging to different OGGs are depicted by *arrowheads* of different colors (intact genes, *filled*; pseudogenes, *outlined*)[5]. The transcriptional orientation of each OR gene is indicated by the orientation of its *arrowhead*

(Carnivora), cow (Cetartiodactyla), and African elephant (Proboscidea). Figure 2b shows the genomic maps of the junctional regions between α- and β-class I ORs in mouse as well as seven placental mammalian genomes deduced from the whole-genome alignments in the UCSC database. This locus comprised a syntenic region among those placental mammals. Genome sequence analysis revealed that a 1.9-kb region that included a cluster of high homology peaks in the middle of the 13-kb region was well conserved among all tested placental mammals (Fig. 2b, Supplementary Fig. 2). Although a β-class I OR gene is pseudogenized in humans, the 1.9-kb region was also conserved in humans, suggesting that it may function as an enhancer not only for β-class I ORs but also for α-class I ORs. Because this conserved 1.9-kb sequence is located in the junctional region of two phylogenetic groups, the α- and β-class I OR genes, we named it the "J element".

To evaluate the importance of the J element, we compared the evolutionary conservation of the J element with that of 35 other mouse OR enhancers, including putative ones[14] (see Methods section). Among the 36 elements examined, 15 elements, including the J element, are conserved in all tested placental mammalian genomes and linked to OR gene (Supplementary Fig. 3). Interestingly, each OR gene linked to the conserved enhancer element belongs to the same orthologous gene group (OGG) among mammalian species (Supplementary Data 1)[5], suggesting that the evolutionarily conserved OR enhancer elements regulate the expression of an OR subfamily whose physiological functions are generally important for mammalian species. Our conservation analysis indicated that the J element was one of the most highly conserved of the 15 conserved OR enhancer elements (Supplementary Fig. 3b). These results strongly suggest that the J element is evolutionarily conserved as an enhancer for class I OR genes in placental mammals.

**Functional analysis of the J element**. To examine the function of the J element, we constructed two transgenes, J13k-gVenus Tg and J-gVenus Tg, in which the 13-kb fragment (Fig. 1c) and the 3.8-kb NcoI fragment including the J element were placed upstream of the 0.9-kb *Olfr544* promoter region and the *gapVenus* reporter gene, respectively (Fig. 3a). Note that the *Olfr544* promoter region itself could not activate reporter gene expression (see Tg-U16, Fig. 1c). Both transgenic (Tg) lines/founders of J13k-gVenus Tg and J-gVenus Tg exhibited reproducible and robust expression of gapVenus in OSNs located in the dorsal MOE, where class I OSNs reside (Fig. 3b). In addition, the majority of gapVenus-positive axons in both Tg mice projected diffusely to multiple glomeruli in the dorsal OB, corresponding to the class I OSN projection domain. These results suggest that the J element possesses class I OSN-specific enhancer activity.

To confirm the class I OSN-specific enhancer activity, we analyzed the co-expression of gapVenus with class I or class II OR genes in J-gVenus Tg mice by two-color in situ hybridization (ISH) (Fig. 3c). GapVenus signals are co-localized with both α- and β-class I genes, suggesting that the J element functions as an enhancer not only for β-class I but also for α-class I genes. In addition, quantification analysis of a total of 6364 neurons from three independent mice demonstrates that gapVenus-expressing OSNs predominantly labeled with class I genes but not with class II genes (class I, 258/3112 (coexpression rate = 8.3%); class II, 4/3252 (0.12%); Fisher's exact test, $p < 2 \times 10^{-16}$; Fig. 3d), indicating that the J element has class I OSN-specific enhancer activity. We also examined the enhancer activity of the J element in zebrafish because some class II enhancers support reporter expression in zebrafish OSNs[14]. However, unlike the H and Lipsi elements, the J element did not activate reporter expression in the OE of any of the tested zebrafish (Supplementary Fig. 1b, c).

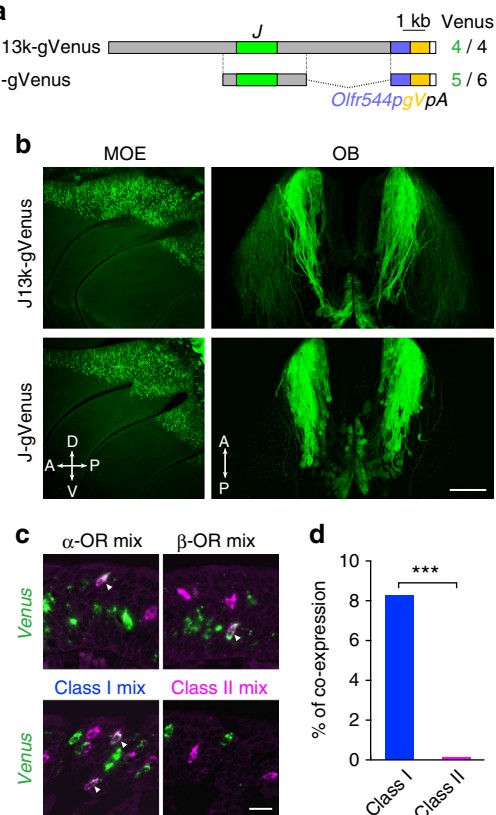

**Fig. 3** The J element possesses class I OSN-specific enhancer activity. **a** Transgenes of J13k-gVenus and J-gVenus. The number of Venus-positive independent founders and lines among the total analyzed is shown on the right. *Olfr544p*, promoter region of *Olfr544*; *gV*, membrane-targeted *Venus* reporter (*gapVenus*); *pA*, polyadenylation sequence. **b** Whole-mount fluorescent images of the MOE and the dorsal OB of hemizygous J13k-gVenus Tg mice and J-gVenus Tg mice. The *scale bar* is 500 μm. **c** Confocal images of two-color ISH of *Venus* (*green*) with mixed probes of α=−class I OR genes, β=class I OR genes, class I OR genes, and class II OR genes (*magenta*) in a J-gVenus Tg mice. Enhancer activity of the J element (*Venus*) was observed in class I OR-positive cells (*arrowheads*). The *scale bar* is 20 μm. **d** Bar graphs showing the percentage of *Venus*-positive cells that are labeled with the class I or class II OR mixed probes (3112 cells for class I, 3252 cells for class II probes from 3 mice). Co-expression preferentially occurred in class I OR-expressing OSNs. Fisher's exact test; $p < 2.2 \times 10^{-16}$, ***$p < 0.001$

**The J element functions as an enhancer for class I OR genes**. To validate the enhancer function of the J element, we generated J element-deleted mice using the CRISPR-Cas9 system (Fig. 4a, b)[20]. Mice homozygous for the J element deletion (ΔJ mice) developed normally and were healthy and fertile.

To comprehensively analyze the effects of ΔJ on all the OR genes, we directly compared the gene expression profiles of ΔJ and littermate wild-type mice using exon microarray analysis (Fig. 4c and Supplementary Data 2). Of 129 intact class I OR genes analyzed on the microarray, the mRNA levels of 75 genes, including three β-class I and 72 α-class I genes, were significantly decreased in ΔJ mice (fold change < −1.3, $p < 0.05$), whereas the mRNA level of only one α-class I gene, *Olrf653*, located 2.1 Mb from the J element, was increased (fold change > 1.3, $p < 0.05$). The mRNA levels of ten class I genes tended to decrease (fold change < −1.3, $p > 0.05$), and those of 43 genes including two atypical class I OR genes, *Olfr556* and *Olfr686*[16], which are

exceptionally expressed in the ventral MOE were unaltered. Although ΔJ had a slightly graded effect on the expression of class I OR genes that was commensurate with genomic distance, i.e., it decreased the expression of more genes located in proximal positions to the J element (log2-fold change vs. genomic distance from the J element; Pearson correlation coefficient $r = 0.42$, $p = 9.7 \times 10^{-7}$), ΔJ also affected the expression of *Olfr692*, which is located at the opposite end of the cluster (2.9 Mb from the J element), indicating that the J element exerts an effect over a ~ 3-Mb genomic region that includes the whole cluster. Because there was no significant effect on class II gene expression or on the expression of any of the tested genes other than ORs, the effect of ΔJ was specific to the class I cluster (Fig. 4d and Supplementary Fig. 4).

**The J element regulates the probability of OR gene choice**. To examine whether the J element regulates the probability of OR gene choice or the level of OR transcripts per OSN, we counted the number of OSNs expressing OR genes in ΔJ and wild-type and (Fig. 4e and Supplementary Table 1). In accordance with the microarray results, the expression of *Olfr544*, *Olfr552*, and *Olfr657* was nearly abolished. The number of opposite-end class I gene *Olfr692*-expressing OSNs was significantly decreased in ΔJ, whereas in *Olfr653* it was significantly increased. There were no obvious differences in the numbers of OSNs expressing atypical class I gene *Olfr556* and the dorsal class II genes *Olfr151* and *Olfr521*. Because the changes in the number of OSNs that express linked class I genes between ΔJ and wild-type mice are in close agreement with those in mRNA levels (Pearson correlation coefficient $r = 0.87$, $p = 3.6 \times 10^{-6}$; Fig. 4f), the J element, such as the H and P elements[12], regulates the probability of OR gene choice.

A massive reduction of the expression levels of class I mRNAs in ΔJ mice indicates that the number of class I genes expressing OSNs is decreased. To investigate the fate of the decreased class I OSNs in ΔJ mice, we quantified apoptotic cells in the dorsal MOE using anti-active caspase-3 antibody. There was no significant difference in the number of apoptotic cells between ΔJ and wild-type mice ($18.5 \pm 4.31$, $n = 3$ for wild-type; $19.2 \pm 3.97$, $n = 3$ for ΔJ; mean ± s.e.m.; unpaired $t$-test, $p = 0.91$), suggesting that the deletion of the J element does not result in cell death. Because there was no significant effect of ΔJ on class II genes expression, it is conceivable that OSNs that fail to express class I genes by ΔJ arrest neuronal differentiation rather express class II genes as secondary choice. Although the decreased number of class I OSNs might result in a thinner dorsal MOE, there was no morphological difference in the dorsal MOE between ΔJ and wild-type mice. Because OSNs expressing class I and class II genes are intermingled within the dorsal MOE, the reduction in the number of class I OSNs could not be detected morphologically.

**The J element operates in *cis***. Next, we examined the *cis*-regulatory effects of the J elements on linked class I genes. By crossing Olfr545-IRES-tauGFP homozygous mice[21] with ΔJ heterozygous (J/ΔJ) mice, we generated two different Olfr545-IRES-tauGFP heterozygous mice in which the GFP-tagged *Olfr545* allele carries an intact J and the GFP-untagged allele carries either an intact J or a mutant ΔJ (Fig. 5a). In Olfr545-IRES-tauGFP heterozygous mice with an intact J (J/J), the co-expression rate of *GFP* with *Olfr545* was 54.3% ($n = 3$), supporting monoallelic and mutually exclusive expression of the tagged and untagged alleles (Fig. 5b). In ΔJ heterozygous mice (J/ΔJ), all the *Olfr545*-positive cells were *GFP*-positive (co-expression rate of 100%, $n = 3$), indicating that deletion of the

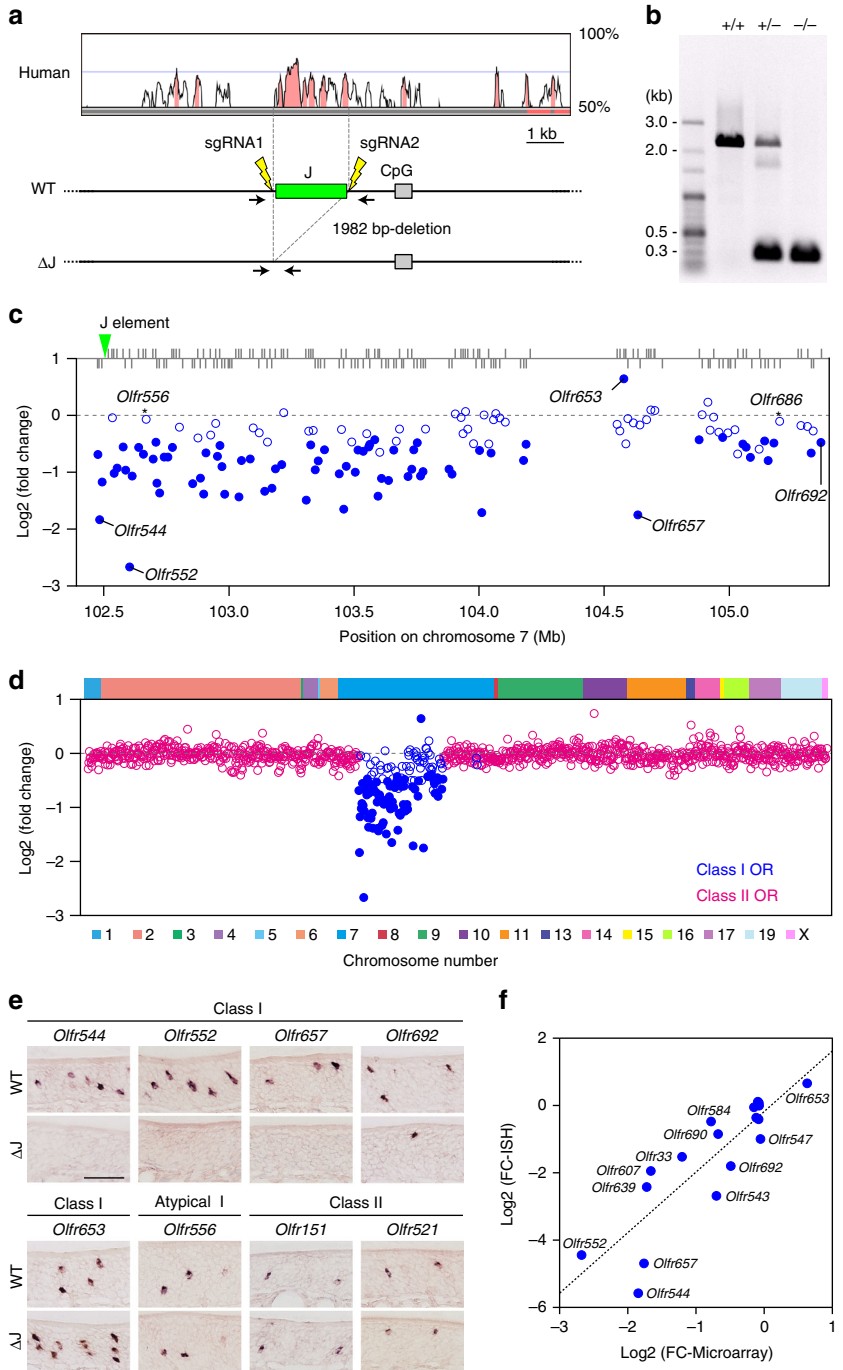

**Fig. 4** Massive reduction in class I gene expression in ΔJ. **a** A targeted deletion of the J element using the CRISPR-Cas9 system. **b** PCR genotyping of wild-type ( + / + ), heterozygous ( + /−), and homozygous (−/−) mice using primers covering the J element. **c, d** Microarray analysis of OR gene expression in ΔJ mice. **c** The log2-fold change (FC) values for class I OR genes are shown in the class I OR cluster position. A fold change criterion of 1.3 (decrease or increase) was used. Differentially expressed class I OR genes ($p < 0.05$) are represented by *filled circles* ($n = 6$). Among 129 tested intact class I OR genes within the class I cluster, the expression of 75 class I OR genes is decreased and that of one gene is increased. Pearson correlation coefficient $r = 0.42$, $p = 9.7 \times 10^{-7}$. **d** The log2 FC values for all tested intact OR genes arranged according to their relative positions along the chromosome (class I, *blue*; class II, *magenta*; $n = 6$). **e** Single-color ISH analysis for five class I OR genes, one atypical class I OR gene, and two dorsal class II OR genes in ΔJ and wild-type mice. The *scale bar* is 50 μm. **f** Correlation between log2 FC values detected by microarray (*horizontal axis*) and ISH (*vertical axis*) for twelve class I, two atypical class I, and four class II OR genes. $n \geq 3$. Pearson correlation coefficient $r = 0.87$, $p = 3.6 \times 10^{-6}$

J element abolished expression of *Olfr545* in *cis*. The intact J allele could not rescue the expression of *Olfr545* of the mutant ΔJ allele in *trans*; thus, the J element operates in *cis*.

**Allelic exclusion depends on the activity of the J element.** Each OSN expresses only one OR gene in a monoallelic and mutually exclusive manner. If allelic exclusion of the OR gene occurs in a stochastic manner at the gene level, in theory each allele is expressed in ~ 50% of the OSNs that express a given OR gene. Indeed, in ΔH and ΔP heterozygous mutant mice, the expression frequencies of class II genes linked to these elements were reduced to ~ 50% of those of wild-type mice[10–12]. We asked

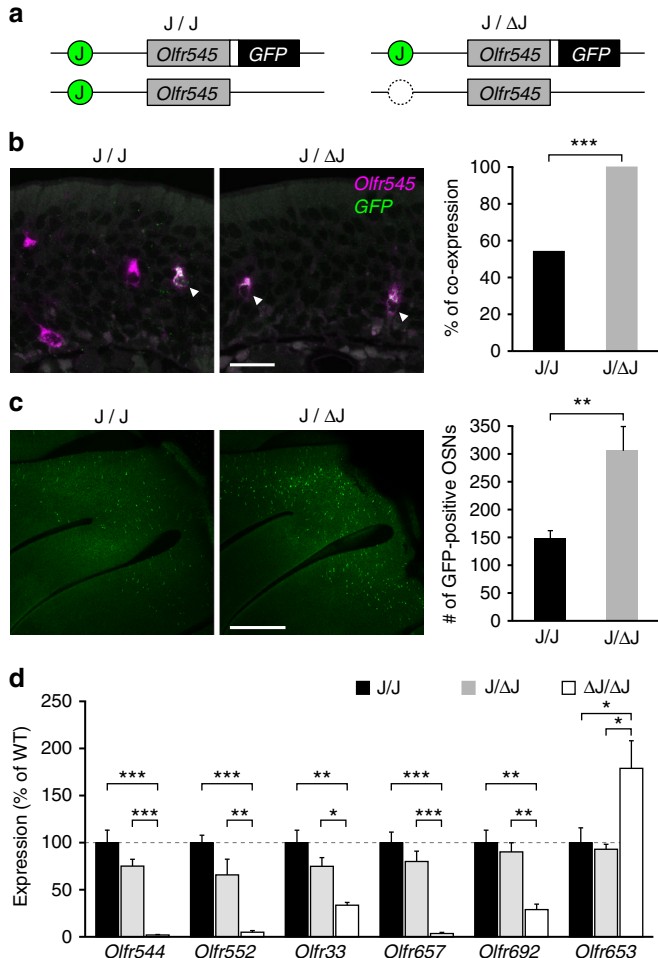

**Fig. 5** Deletion of the *trans* J element increases expression frequency from the intact J allele. **a** Schematic illustrations of the *Olfr545* alleles of mice used in **b** and **c**. One allele contains the intact J element and tauGFP-tagged *Olfr545*, and another allele contains the intact or deleted J element and untagged *Olfr545*. The *white box* represents the *IRES* sequence. **b** Confocal images of two-color ISH for *Olfr545* (*magenta*) and *GFP* (*green*). The *bar graph* shows the percentage of co-labeled cells: 124 co-labeled cells and 107 *Olfr545* single-positive cells in J/J (54,3%, from 3 mice), 213 co-labeled cells and no *Olfr545* single-positive cells in J/ΔJ mouse (100%, from 3 mice). The intact J element of the GFP-tagged wild-type allele did not rescue the defective phenotype of *Olfr545* expression from the mutant allele. The *scale bar* is 20 μm. Fisher's exact test; $p < 2.2 \times 10^{-16}$, ***$p < 0.001$. **c** Whole-mount confocal images of the turbinate. The *bar graph* shows the numbers of GFP-positive OSNs on the medial face of the turbinate in half-head. The number of GFP-labeled OSNs of J/ΔJ is ~ 2 times larger than in J/J mice. The *error bars* represent the mean ± s.e.m. (*n* = 6 for J/J, *n* = 5 for J/ΔJ). The *scale bar* is 500 μm. Unpaired *t*-test, *p* = 0.0046, **$p < 0.01$. **d** Percentage of OSNs expressing class I genes in ΔJ heterozygous (J/ΔJ) or homozygous (ΔJ/ΔJ) mice compared to wild-type (J/J, set as 100%). The *error bars* represent the mean ± s.e.m. (*n* ≥ 4). One-way ANOVA in each *Olfr* probes. Tukey's post hoc multiple comparisons test shows significant difference between indicated conditions. *$p < 0.05$, **$p < 0.01$, ***$p < 0.001$. Quantification data are summarized in Supplementary Table 2

whether a similar effect of the ΔJ heterozygous mutation on the expression frequency of class I genes is observed. We quantified the number of GFP-positive OSNs in Olfr545-IRES-tauGFP heterozygous mice with either an intact J (J/J) or a heterozygous ΔJ mutation (J/ΔJ) background in whole-mount specimens

(Fig. 5a–c). Unexpectedly, the number of GFP-positive cells in J/ΔJ was nearly double that in J/J (148 ± 13.8, *n* = 6 for J/J; 306 ± 43.5, *n* = 5 for J/ΔJ; mean ± s.e.m.), indicating that expression of the GFP-tagged *Olfr545* allele with the intact J increased two-fold in J/ΔJ.

To further confirm this observation, we quantified the number of OSNs expressing other class I OR genes by ISH. If the expression frequency from the intact J allele did not increase in J/ΔJ, the number of OSNs expressing class I OR genes would be reduced to ~ 50% of the sum of the number of wild-type (J/J) and ΔJ homozygous mice (ΔJ/ΔJ). However, there was no significant difference in the number of class I genes expressing OSNs in J/J and J/ΔJ mice (Fig. 5d and Supplementary Table 2). These results support the idea that the frequency of class I gene expression from the intact J allele increases to complement the loss or reduction of class I gene expression from the ΔJ allele. In other words, the allelic exclusion of class I genes is not solely established by a stochastic choice of a single class I allele but depends on the activity of the J element.

**Conserved sequence motifs in the J element**. Finally, we analyzed the conserved sequence motifs in the J element. In the class II enhancers, the H and P elements contain conserved sequence motifs of multiple homeodomain sites and neighboring O/E-like sites that are essential for their enhancer activities[10, 22, 23]. To examine conserved sequence motifs in the J element, we scanned the mouse J sequence and found several homeodomain sites and O/E-like sites (Fig. 6a). Intriguingly, the homeodomain and O/E-like sites were clustered within the 430-bp region that corresponded to the highest homology region between the mouse and human J elements (hereinafter referred to as the core J element). This motif organization of multiple homeodomain sites and a neighboring O/E-like site is similar to that found in the H and P elements (Fig. 6a), suggesting that the sequence motifs in the J element also have a critical role in class I OR gene expression.

Because class I ORs exist not only in placental mammals but also in marsupials and monotremes, it is possible that functional motifs in the core J element are conserved in all mammals from humans to the platypus. By focusing on the 430-bp core J element, we further analyzed conserved sequence motifs among mammals by constructing hidden Markov models (HMM). Using HMM, the top hit sequence was obtained from four aplacental mammalian genomes, including three marsupials (opossum, wallaby, and Tasmanian devil) and a monotreme (platypus). The obtained genomic sequences were orthologous to the core J element as shown by the fact that they resided near the β-class I OR genes, *Stim1* and *Rrm1* as in the placental mammals. Alignment of the core J element sequence in eleven mammalian species revealed following conserved motifs (Fig. 6b), four homeodomain sites, a neighboring O/E-like site, and a conserved sequence motif (5′-AAACTTTTC-3′). The 5′-AAACTTTTC-3′ motif that was perfectly conserved among the analyzed mammalian genomes was not present in the H and P elements but was specific to the J element, suggesting that this motif may be responsible for the characteristic features of class I gene expression. Identification of factors that bind to the class I-specific conserved sequence motifs will further unveil molecular mechanisms underlying J-element dependent class I OR gene expression.

## Discussion

The present study identified a *cis*-regulatory element for class I OR genes. Compared with the deletions of class II OR enhancers of ΔH, ΔP, and ΔLipsi, which affect the expression of 7–10 class

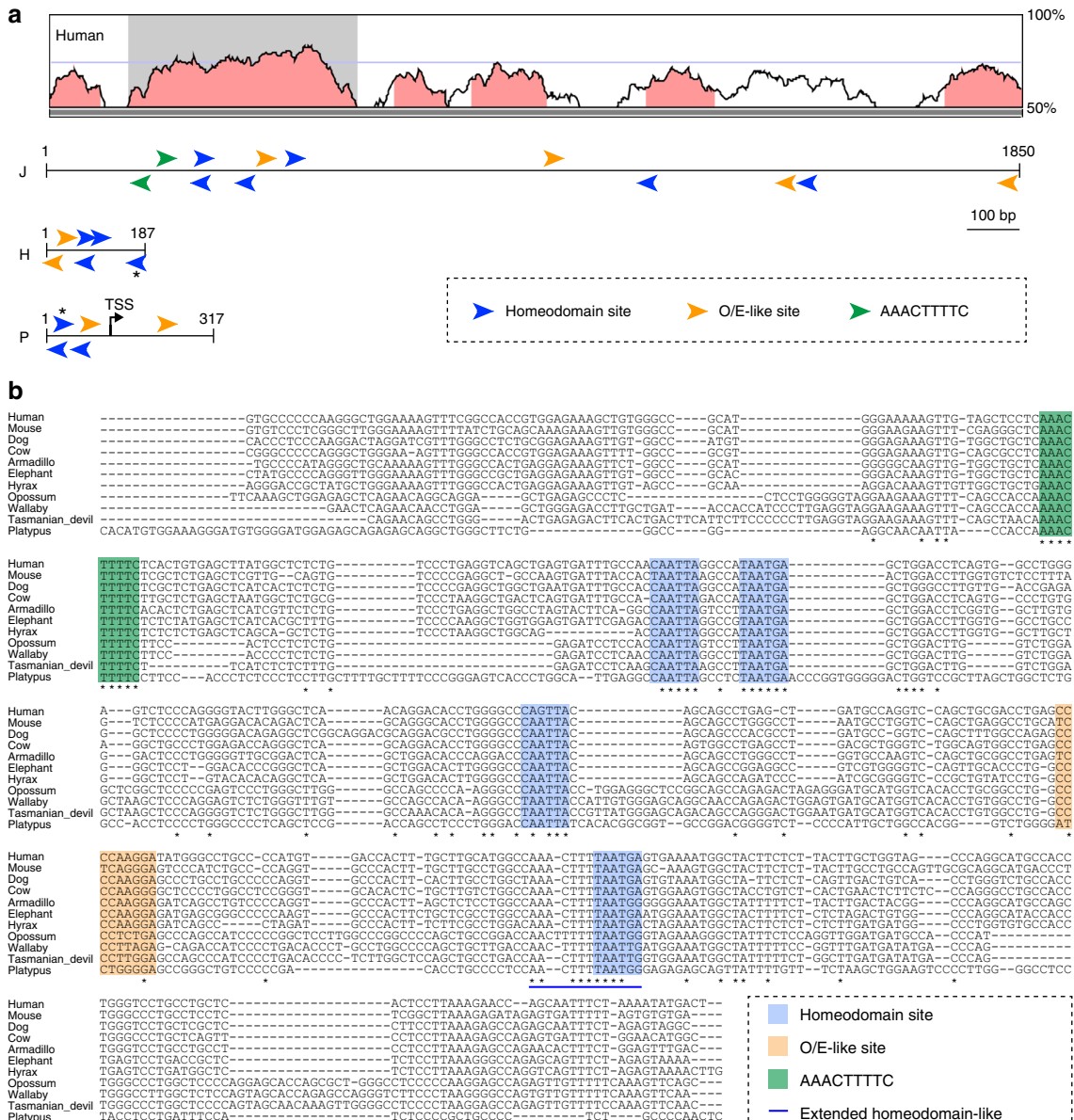

**Fig. 6** Conserved sequence motifs in the J element. **a** Schematic illustration of putative binding motifs of the homeodomain site (TAATKR; *blue arrowheads*), an O/E-like site (YYYCARRRR; *orange arrowheads*), and a class I-specific conserved motif, the AAACTTTTC site (*green arrowheads*) in the J, H, and P elements in mice. The *asterisks* represent the extended homeodomain site (5′-AACTTTTTAATGA-3′). The orientation of the motifs on either DNA strand is represented by arrowheads pointing in different directions. The VISTA plot between mouse and human is shown at the *top*. The *shaded box* represents the core J element with highest homology; it contains multiple conserved motifs. **b** Multiple alignment of a 430-bp sequence, the core J element (*shaded box* in **a**), from eleven mammals, including placental and aplacental mammals. Comparison of the sequences reveals six stretches of homology, which are indicated by *colored boxes*: one AAACTTTTC sequence (*green box*), four potential homeodomain sites (*blue*), and one O/E-like site (*orange box*). The *blue line* depicts a stretch of additional sequence homology upstream of the homeodomain called the extended homeodomain-like site

II genes within a ~ 200 kb genomic range[10–12, 14], the impact of ∆J is much larger both in terms of number of genes (75 genes) and genomic reach (~ 3 Mb). Moreover, the J element regulates the class I gene expression of much larger number of genes and over a much greater genomic distance than any other known enhancer elements, e.g., the largest number was ~ 30 genes of the cluster control region for the protocadherin-β cluster and some protocadherin-γ genes[24, 25], and the longest genomic distance was ~ 1.3 Mb of 3′ enhancer element for the protooncogene *Myc*[26]. Thus, our experiments demonstrate the existence of a long-range enhancer element for the class I cluster. The mechanisms underlying the differential impact of the enhancer activity of the

J element and the class II enhancers are unknown. One possible explanation for a long-range regulation of the J element is that class I genes throughout the cluster can be accessible by the J element, which is established by the organization of 3D chromatin structure of this locus. The probability of class I OR gene choice depends on 3D chromatin structure rather than linear genomic distance.

The J element was not listed in the original catalog of 35 OR enhancers that were identified based on the commonality of specific epigenetic signatures associated with the H element[14]. Therefore, the J element may have epigenetic modifications that differ from those of class II enhancer elements. Indeed, meta

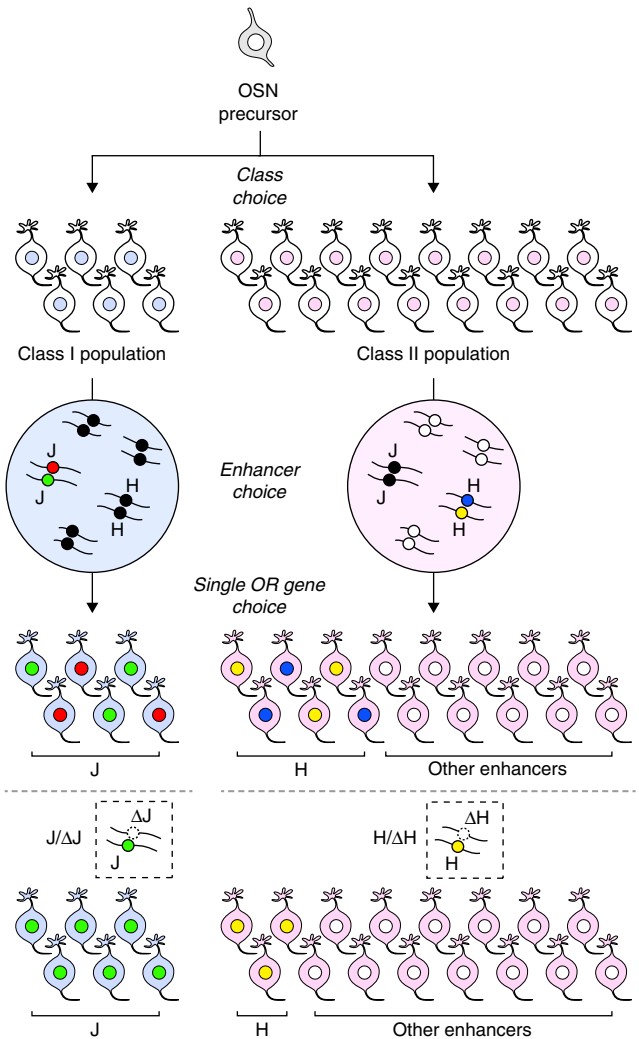

**Fig. 7** Model for the enhancer-dependent process of the singular OR gene expression. OSNs are fated to express a single OR gene from the repertoire of class I or class II genes to produce fixed populations of class I (*blue*) and class II OSNs (*magenta*). Subsequently, a functional OR enhancer is chosen according to the OSN fate to open up local chromatin structure to orchestrate the first step of OR gene choice. Class I OSNs activate one enhancer allele of the J element (*green* or *red circle*) or of a limited number of enhancer elements. By contrast, class II OSNs select one element from the class II enhancer repertoire spread over most chromosomes (*white circle*, H element is colored in *yellow* or *blue*). The selected enhancer element chooses one functional OR allele to express. In ΔJ heterozygous mutation, the intact J element allele (*green*) is activated to complement the ΔJ allele to fill in the class I OSN population. In class II enhancer (e.g., ΔH) heterozygous mutation, an intact element is selected randomly from the class II enhancer repertoire spread over chromosomes. In consequence, the expression of the H element linked OR genes are reduced to ~ 50%

analysis of the sequencing data of GSE55174 and GSE52464 (Gene Expression Omnibus) used to identify 35 OR enhancer elements[14] reveals that epigenetic landscape of the J element is different from previously characterized enhancer elements for class II genes (Supplementary Fig. 5). In addition, the level of constitutive heterochromatin (H3K9me3 and H4K20me3) is lower at class I loci than at class II loci[6]. These observations suggest that differential epigenetic modifications of both the J element and class I OR loci may contribute to the long-range enhancer activity of the J element.

Although ΔJ results in a massive reduction in class I OR gene expression, the extent of reduction of gene expression in ΔJ differed among the cluster, and the expression of 43 class I genes was not affected. In addition, an increase in the expression of one gene was observed in ΔJ. These results suggest that the J element is not solely responsible for class I OR gene expression and that at least one additional enhancer element for the class I locus exists. One of the candidates for such an additional enhancer is the Milos element listed in the catalog of OR enhancers, which resides inside the class I cluster located 0.9 kb telomerically from the J element. Although its conservation index is low, the Milos element is conserved in all placental mammals (Supplementary Fig. 3b). Alternatively, because ΔJ has a slightly graded effect on the expression of class I OR genes commensurate with genomic distance, an additional enhancer element may be located on the telomeric side of the cluster.

During evolution, class I genes have been retained in a single cluster on a single chromosome in all mammalian species analyzed to date. Why the class I genes did not migrate outside of the cluster has remained a mystery in OR evolution. Because the J element regulates most class I genes at the cluster level, we propose a hypothetical model that accounts for the single class I cluster in which a single or a limited number of enhancer elements governs the expression of class I OR genes at the cluster level. Because duplication and translocation of a given gene without a *cis*-regulatory element would result in failure of that gene's expression, the presence of such an element might suppress the duplication/translocation of class I OR genes to positions outside the cluster. In accordance, previous genome sequence analysis demonstrated that class I genes show a significantly lower evolutionary rate and a significantly smaller number of gene gains and losses than class II OR genes[5].

Interestingly, two class I genes, *Olfr503* and *Olfr504*, are exceptionally identified outside the class I cluster only in the mouse genome (Supplementary Fig. 6a)[13]. The *Olfr503* and *Olfr504* genes reside tandemly in the class II cluster of P-telomeric, which is 3 Mb from the telomeric end of the class I cluster. Because of their extremely high nucleotide identities to two neighboring class I genes, *Olfr658* and *Olfr657* (*Olfr503*: 97% identical to *Olfr658*; *Olfr504*: 96% identical to *Olfr657*), the *Olfr503* and *Olfr504* genes were probably duplicated from the class I cluster. However, the expression of *Olfr503* and *Olfr504* was not detected or barely detected in MOE transcriptome analysis (FPKM = 0.00–0.03)[27] and in RT-PCR analysis (Supplementary Fig. 6b), suggesting ongoing pseudogenization of these genes. These observations support our hypothesis that class I genes must reside within the cluster for their proper expression.

Our genetic experiments revealed that the J element operates in *cis* in a similar manner to the H and P elements. In Δ*H* and Δ*P* heterozygous mutant mice, the expression of class II genes linked to these elements was reduced to ~ 50% of that of wild-type mice[10–12]. By sharp contrast, the deletion of the *trans* J element increased the expression frequencies of class I genes from the intact J allele. Our interpretation of this interesting phenomenon is that there is a fixed population of OSNs that restricted to choosing the class I repertoire prior to first choice (Fig. 7). For the class I OSN population, the expression of class I genes is regulated by the J element or by a limited number of enhancer elements including the J element. The phenotype of the ΔJ heterozygous mutation is complemented by the biased choice of intact J element and/or a limited number of functionally related elements to fill in the class I population. In contrast, there exist many enhancer elements in the class II OSNs population. ΔH and ΔP heterozygous mutations are probably complemented by alternative choice of other functional enhancers from the repertoire of class II enhancers spread over almost all

chromosomes and thereby caused to express other class II genes to fill in the class II population (Fig. 7). In this scenario, the intact H or P element allele can become active, but its probability of choice is not biased by dilution of the choice of other class II enhancers.

A similar phenomenon was reported in gene-targeted mice with a cluster deletion of trace amine–associated receptor (TAAR) genes; in these animals, the number of neurons expressing a given TAAR gene doubled when one copy of the TAAR cluster was removed[28]. The authors argued that there is a fixed population of neurons that is restricted to express TAARs. Although it is not known whether the cluster deletion of TAAR genes also deletes their enhancer element/elements, our results clearly demonstrate that singular class I gene expression is not solely established by the stochastic choice of a single class I allele; rather, it orchestrates with the activity of the linked enhancer element. It is conceivable that the allelic exclusion observed for class I genes is simply a result of the J-dependent activation of a single functional class I allele. Therefore, we propose a model of the enhancer-dependent process that results in singular OR gene expression in which (1) OSNs are fated to express a single OR gene from the repertoire of class I or class II genes[21] to produce fixed populations of class I and class II OSNs; (2) a functional enhancer element is chosen according to the OSN fate in a stochastic and MOE region-specific manner; and (3) a selected enhancer element chooses a single OR gene to express (Fig. 7). Our model is compatible with the model proposed by Magklara et al. in which the singular OR gene is chosen by stochastic escape from heterochromatic silencing[6, 29], if the process of stochastic escape should orchestrate with the choice of a functional enhancer element.

## Methods

**Mice**. All mice were housed in standard conditions with a 12 h light/dark cycle, and access to food and water ad libitum. Mutant and wild-type mice at 4–6 weeks old of either sex were used for analyses. All mouse studies were approved by the Institutional Animal Experiment Committee of the Tokyo Institute of Technology, and were performed in accordance with institutional and governmental guidelines. Olfr545-IRES-tauGFP mice were generated as described previously[21].

Transgenic mice carrying Tg-220 and its deletion series were generated with a standard pronuclear microinjection protocol[30]. The purified transgenes were microinjected into the pronucleus of B6C3F1 (C3H/HeSlc male × C57BL/6NCrSlc female, Japan SLC, Inc.) mouse zygotes. To obtain B6C3F1 zygotes, over 4 weeks old B6C3F1 female mice were treated with superovulation, and then mated with adult B6C3F1 male mice. Olympus IX-71 microscope equipped with the micromanipulator transgenic system (Narishige) and FemtoJet system (Eppendorf) were used for microinjection. Injected eggs were transferred to the oviducts of pseudopregnant female ICRs (over 6 weeks old, Japan SLC, Inc.). The founders were screened by PCR with the three primer sets that specifically amplified the internal (EGFP) and left (SacB) and right (CmR) ends[17] (Supplementary Table 3).

For the J13k-gVenus and J-gVenus Tg mice, the Tol2 cytoplasmic microinjection method[30] was used. Briefly, the Tol2 transgene plasmids were purified by phenol-chloroform extraction and ethanol precipitation, and then dissolved a TG buffer (10 mM Tris-HCl of pH 8.0; 0.1 mM EDTA) prepared with DEPC-treated water. Tol2 transposase mRNA was synthesized in vitro using the NotI-linearized pCS-TP with the mMESSAGE mMACHINE SP6 Kit (Thermo Fisher Scientific), in accordance with the manufacturer's instructions. DNA solution containing 20 ng per µl circular Tol2-trangene plasmid and 25 ng per µl transposase mRNA was injected into the cytoplasm of fertilized eggs. To obtain F1 mice and establish Tg lines, some founder mice were crossed with adult C57BL/6NCrSlc mice (Japan SLC, Inc.). J13k-gVenus Tg line #14 and J-gVenus Tg line #17 were used.

The J element was deleted using the CRISPR-Cas9 genome editing technique with a circular plasmid injection method[20] using pX330 vector (Addgene# 42230)[31]. Two gRNAs were designed at upstream and downstream of the J element, using Zifit (http://zifit.partners.org/ZiFiT/) and blastn programs to reduce potential off-targets. The oligonucleotide sequences for gRNAs were 5′-CACCGTCTCATTGC CACCCGGATGA-3′ (gRNA1 sense), 5′-AAACTCATCCGGGTGGCAATGAG AC-3′ (gRNA1 antisense), 5′-CACCGCCCCTCCACCGTACTTGCA-3′ (gRNA2 sense), and 5′-AAACTGCAAGTACGGTGGAGGGGC-3′ (gRNA2 antisense). The plasmids expressing hCas9 and sgRNA were prepared by ligating oligos into BbsI site of pX330. For the targeted deletion of the J element, the two pX330 plasmids mixture (2.5 ng per µl each) were coinjected into pronuclei of

B6C3F1 fertilized eggs by the same method generating Tg mice. By screening the deletion allele by PCR and DNA sequencing using primers flanking J region (Supplementary Table 3), we obtained three independent founder mice from 19 candidate mice. All potential off-target sites identified as exactly 12 bases matches at the 3′ end of gRNA sequences with the PAM sequence (NGG) were analyzed by direct sequencing of genomic PCR products, and confirmed that there were no indel mutations (Supplementary Table 4). We backcrossed one founder to C57BL/6NCrSlc mice (Japan SLC, Inc.), and established ΔJ mice line #8 with a 1982 bp full deletion (mm10, chr7: 102,509,690 - 102,511,671) of the J element. Second backcross generation of ΔJ heterozygous mice were used for the microarray analysis, and second and third backcross generations were used for ISH.

**Transgene construction**. Deletion of the upstream region of Tg-220 was performed in the B. subtilis genome (BGM) vector system[17]. The BAC plasmid p108BGMC containing a cat gene for B. subtilis was used to construct upstream deletion plasmids. Homology arms were amplified by KOD plus DNA polymerase (Toyobo) using the RP23-61O11 BAC clone as a template. The 1.2 kb homology arms homologous to -79 (Tg-U79), -29 (Tg-U29) or -16 kb (Tg-U16) position of Tg-220 were cloned into p108BGMC as HindIII–BamHI fragments. All deletion plasmids were linearized with BamHI, and used for transforming the BGM clone carrying Tg-220. The transformants were screened with chloramphenicol (5 µg per ml). Transformation of B. subtilis and purification of the transgenes were performed as described elsewhere[17]. Briefly, B. subtilis was routinely grown in 1–5 ml Luria-Bertani (LB) broth at 37 °C by rotating ( > 50 r.p.m.) or shaking (200 r.p.m.). Antibiotic Medium 3 (Difco) was used for the selection of the BGM transformants. For transgene extraction, genomic DNA carrying the transgene in an agarose plug was prepared, digested with I-PpoI and resolved by contour-clamped homologous electric field gel electrophoresis in a 1% (w/v) agarose gel using sterile 0.5 × TBE. The separated transgene band was concentrated in 4% (w/v) agarose by conventional electrophoresis with 0.5 × TBE. The excised concentrated transgene band was placed in a prepared dialysis tube hydrated with 0.5 × TBE, and the transgene was electroeluted. The eluate was dialyzed with injection buffer (10 mM Tris–HCl (pH 8.0), 0.1 mM EDTA, 100 mM NaCl) at 4–6 °C overnight. Primer sequences are summarized in Supplementary Table 3.

To construct transgenes for J13k-gVenus and J-gVenus, the 2.4 kb fragment homologous to the 5' end and the 0.9 kb fragment homologous to the 3' end of the 13 kb region were subcloned into pBluescript II SK(+) vector (Stratagene) as SalI-BamHI and BamHI-XbaI fragments, respectively. Both fragments have an endogenous PacI site. The 10 kb PacI fragment from the RP23-61O11 BAC clone was inserted into the subcloned plasmid to generate the full-length 13 kb genomic region as a SalI-XbaI fragment. The 13 kb fragment and the 3.8 kb NcoI fragment including the J element were inserted into a reporter vector, which is a modified pBluescript II SK(+) vector containing the gapVenus-pA fragment and the PCR-amplified Olfr544 promoter fragment (~ 870 bp upstream of TSS plus ~ 40 bp of noncoding exon). For Tol2-transposon mediated transgenesis, the transgenes were transferred to a Tol2 donor plasmid, pTol2-EPA vector, which was constructed by inserting a linker containing EcoRI, AscI and PacI sites between SalI and BglII sites of the HSF51 vector (provided from Dr. H. Nishihara). The Olfr544 promoter, the 5'-end and 3'-end fragments of the 13 kb region were amplified by PrimeSTAR HS DNA polymerase (TaKaRa) using the BAC clone as a template.

**Transient reporter assay in zebrafish**. Transient reporter assay in zebrafish was performed as described previously[32]. To generate J-hspGFF driver construct, a PCR-amplified 1.9 kb J element was cloned into pT2KhspGFF modified vector, which contains hsp basal promoter a modified GAL4 transcription activator and Tol2 transposon elements. DNA solution containing 20 ng per µl J-hspGFF plasmid DNA and 20 ng per µl Tol2 transposase mRNA was injected into one-cell-stage fertilized eggs of UAS:GFP line (kindly provided by Dr. K. Kawakami). Bright field and GFP fluorescence images were taken with Leica FLUOII fluorescent stereomicroscope.

**Nucleotide percent identity plot**. The nucleotide sequence of an approximately 220 kb mouse genomic region of Tg-220 (RP23-61O11, Accession#: AC102535) was compared with those of the class I OR cluster region in other species by using the web-based VISTA program (http://genome.lbl.gov/vista/mvista/submit.shtml)[33]. The Shuffle-LAGAN for the alignment program and the following settings of VISTA parameters were used; conservation identity: 70%; calculation window and minimal conserved width: 100 bp. Because the class I OR gene cluster and non-OR genes (Stim1 and Rrm1 outside the cluster, and Trm21 inside the cluster) compose a syntenic region among mammals, 2 Mb genomic sequences from Stim1 locus to the class I OR gene cluster, which is maximum length to submit, were used to analyze homology among mammalian species. Following sequences were compared; mouse (Mus musculus, mm10) chromosome 7: 102,410,377–102,630,379; human (Homo sapiens, hg38) chromosome 11: 3,855,702–5,855,701; rhesus macaque (Macaca mulatta, rheMac8) chromosome 14: 61,564,147–63,564,146; horse (Equus caballus, equCab2) chromosome 7: 71,934,503–73,934,502; dog (Canis familiaris, canFam3) chromosome 21: 26,394,689–28,394,688; cow (Bos taurus, bosTau8) chromosome 15: 50,049,269–52,049,268. Repetitive elements of the mouse sequence were masked with RepeatMasker on the VISTA server.

For zebrafish (*Danio rerio*, danRer10), a 2 Mb genomic region containing all five β-ORs (*or112*, *or113*, and *or114* subfamilies)[34] was used; chr15: 4,355,001–6,355,000. For frog (*Xenopus tropicalis*, xenTro9.0), a 556 kb scaffold fragment containing all 14 β-ORs[19], one α-OR, and *Rrm1* was used; scaffold_147: 1–555,726. For high-sensitive detection, calculation window and minimal conserved width were set to 50 bp.

**The extent of conservation of the J element.** We compared the nucleotide sequences of a region syntenic to the enhancer elements in another species' genome. The genomic regions from each pairwise alignment were extracted according to the coordinates in the mouse genome for each of the 36 enhancers (the 35 + J elements) for the tested seven placental mammalian species. The following whole-genome pairwise alignments between mouse (mm10) and another species were obtained from the web site of USCS Genome Informatics (http://hgdownload.soe.ucsc.edu/downloads.html): mouse-human (hg38), mouse-guinea pig (cavPor3), mouse-rabbit (oryCun2), mouse-horse (equCab2), mouse-dog (canFam3), mouse-cow (bosTau8), and mouse-elephant (loxAfr3). Because the coordinates of the 35 enhancer elements in Supplementary Table 2 of Markenscoff-Papadimitriou et al. are based on the mouse genome version mm9[14], we first converted them into those based on mm10, and then extracted the genomic regions from each pairwise alignment according to the coordinates in the mouse genome for each of the 36 enhancer elements including the J element. If the syntenic region of a given enhancer element was not found in at least one of the seven pairwise alignments, the enhancer was eliminated from further analysis as a poorly conserved sequence. The coverage is calculated as $A / L$, where $L$ is the length in nucleotides of a given mouse enhancer element and $A$ indicates the number of nucleotides aligned to the other species' genome within the enhancer element. The identity represents the ratio of identical nucleotides (among the aligned nucleotides) between two species.

Next, we asked whether the conserved elements in the non-mouse genomes are also located close to an OR gene/cluster. Here, we considered only enhancer elements that have an OR gene within 1 Mb on the same chromosome or on the same scaffold. For this purpose, we examined the OR gene nearest the conserved elements in each of the seven mammalian genomes for each of the 21 enhancer elements (Supplementary Data 1). To evaluate the level of conservation in terms of both length and identity, we defined the conservation index by multiplying the percentage coverage by the nucleotide identity: (coverage [%] × identity [%])/100.

**Analysis of wholemount specimens.** Fluorescent images of endogenous EGFP and Venus signals in whole-mount specimens were taken with Olympus SZX10 fluorescent stereomicroscope with a DP71 digital CCD camera and Leica SPE confocal microscope. Confocal images were collected as z-stacks and projected into a single image for display. Images were adjusted and merged using Photoshop CC (Adobe). To quantify EGFP-tagged *Olfr545* expressing OSNs in Olfr545-IRES-tauGFP heterozygous mice at 5–6 weeks old, endogenous EGFP fluorescence signals of the medial face of the turbinate in half-head whole-mount specimens were counted.

**In situ hybridization.** Mice were transcardially perfused with 4% paraformaldehyde in PBS. Dissected MOE tissues were postfixed overnight at 4 °C. Tissue was decalcified in 0.45 M EDTA in PBS. After cryoprotection by 15% and 30% sucrose in PBS, tissue samples were embedded in FSC 22 Frozen Section Media (Leica). The MOE was cryosectioned coronally at 12 μm thick.

OR coding sequence was cloned into pGEM-T easy vector (Promega) from the cDNA or genomic DNA. Digoxigenin- (DIG) or dinitrophenyl- (DNP) labeled antisense cRNA probes were synthesized by in vitro transcription with T7, SP6, or T3 RNA polymerase (Roche). Information of all riboprobes is summarized in Supplementary Table 5. Probes for *Olfr17*, *Olfr151*, and *Olfr19* were prepared as described previously[11]. The Venus mRNA was detected by an *EGFP* coding probe. Following mixed probes were used in two-color *in situ* hybridization (ISH); α-class I mix (*Olfr78*, *Olfr672*), β-class I mix (*Olfr543*, *Olfr544*), class I mix (*Olfr78*, *Olfr544*, *Olfr552*, *Olfr578*, *Olfr672*, *Olfr692*,), class II mix (*Olfr19*, *Olfr54*, *Olfr73*, *Olfr151*, *Olfr521*, *Olfr878*).

Single-color and two-color ISH was performed according to methods described previously[35, 36]. Briefly, the post-fixed sections were pre-treated by the following procedures, permeabilization, HCl-treatment and acetylation, and were hybridized with hapten-labeled probes at 65 °C overnight. After hybridization and wash steps, we conducted the chromogenic or fluorescent detections. For the chromogenic detection of single-color ISH, sections were incubated with 1% (v/w) DIG blocking reagent (Roche) and an anti-DIG-AP antibody (Roche, 11093274910, 1/1000 dilution) at room temperature for 60 min each. After washing in TN buffer (100 mM Tris-HCl of pH 7.5; 150 mM NaCl) with 0.01% Tween20, ISH signals were detected by alkaline phosphatase chromogens, nitro blue tetrazolium (NBT) and 5-bromo-4-chloro-3-indolyl-phosphate (BCIP) (Roche) in DIG3 buffer (100 mM Tris-HCl of pH 9.5; 100 mM NaCl; 100 mM MgCl₂; 0.01% Tween20) at room temperature overnight. Chromogenic stained sections were air-dried completely, and mounted with the Entellan New Mounting Medium (Merck Millipore).

For the fluorescent detection of two-color ISH, sections were treated with 0.5% (v/w) TSA blocking reagent (PerkinElmer) in TN buffer with 0.05% Tween20 after the stringent washes, and were incubated with anti-DIG-AP (Roche, 11093274910, 1/1000 dilution) and anti-DNP-HRP (PerkinElmer, FP1129, 1/1000 dilution) antibodies in blocking buffer at 4 °C overnight. DNP-labeled target was detected by a combination of rabbit anti-DNP-KLH (Thermo Fisher Scientific, A-6430, 1/300 dilution) and Donkey Alexa Fluor 488-conjugated anti-rabbit IgG (Jackson ImmunoResearch Laboratories, 711-545-152, 1/500 dilution) antibodies after the amplification by the Tyramide Signal Amplification (TSA) Plus DNP System (PerkinElmer). Subsequently, DIG-labeled target was detected by using the HNPP/FastRed detection kit (Roche). Polyvinyl alcohol mounting medium with DABCO (SIGMA) was used for a fluorescent detection of two-color ISH signals.

Images of ISH signals were taken with Olympus BX51 microscope with a DP71 digital CCD camera and Leica SPE confocal microscope. To quantify the number of OR-expressing OSNs, every 30th coronal section (12 μm thickness) throughout the MOE of wild-type, heterozygous, and homozygous ΔJ mice at 4–5 weeks old was collected. The number of positive cells of five coronal sections from seven sections before the first appearance of the OB was counted. Fold change by ISH was calculated by dividing the average number per sections of ΔJ mice by that of wild-type mice.

**Microarray analysis.** MOEs were obtained from six ΔJ and six wild-type mice at P30 or P33. The total RNA of each preparation was extracted using RNeasy Mini Kit (QIAGEN). Labeling, hybridization, and scanning were performed according to standard Affymetrix protocols by the using the Affymetrix GeneChip Mouse Gene 1.0 ST Array and GeneChip WT PLUS Reagent Kit. The array data were analyzed using GeneSpring GX 14 software, and annotated using MoGene1.0.MoGene-1_0-st-v1_na35 genome data (Agilent Technologies). Data normalization algorithm used 75% percentile shift. The fold change value was calculated by dividing the expression value of ΔJ by that of wild-type. Testing for differences for each olfactory receptor gene in total 1213 olfactory receptor gene probe data was done using Moderated *t*-test with multiple testing correction of Benjamini Houchberg FDR. The expression data were then exported and arranged in Excel for presentation (Supplementary Data 2).

To create a phylogenetic tree of mouse class I ORs, 129 intact class I OR amino acid sequences were aligned by E-INS-i program in MAFFT[37]. From this alignment, a gene tree was generated using the Neighbor-Joining method implemented in MEGA6 software[38].

**Motif analysis.** To analyze motif conservation among mammals, the highest homology sequence was obtained using blastn program (NCBI blast + 2.3) and a 430 bp sequence of the mouse core J element (the highest conserved peak with human in VISTA; chr7:102,509,921–102,510,350) as a query in each six placental mammals; human (hg38), dog (canFam3), cow (bosTau8), armadillo (dasNov3), elephant (loxAfr3) and hyrax (proCap1). A multiple alignment of the sequences was performed by MUSCLE-3.8.31[39], and then used to construct hidden Markov model (HMM) by nhmmer of HMMER-3.1 package[40]. Using the HMM, the top hit sequence was obtained from each four aplacental mammalian genomes; opossum (monDom5), wallaby (macEug2), Tasmanian devil (sarHar1) and platypus (ornAna2). The obtained core J element sequences in 11 mammals were aligned by MUSCLE-3.8.31.

**RT-PCR analysis.** Total RNA was extracted from total MOE as described in microarray analysis. First strand cDNA was synthesized from 0.5 μg RNA with oligo-dT primer using ThermoScript Reverse Transcriptase (Thermo Fisher Scientific). RT-PCR was performed using TaKaRa Ex Taq (TaKaRa) with primers specific for each OR genes. PCR reactions for OR genes were performed at annealing temperature of 62 °C for *Olfr657* and *Olfr504*, 60 °C for *Olfr658*, or 58 °C for *Olfr503*, respectively, with cycling conditions as follows: 94 °C for 2 min, 32 cycles of 10 sec denaturation at 98 °C, 30 sec annealing, and 30 sec extension at 72 °C by 5 min extension at 72 °C. PCR reaction for *Gapdh* was 94 °C for 2 min, 27 cycles of 10 s denaturation at 98 °C, 30 sec annealing at 65 °C, and 54 s extension at 72 °C by 5 min extension at 72 °C. A measure of 0.4 ng genomic DNA was used as a positive control.

**Statistical analysis.** Statistical analyses were performed by R and GraphPad Prism. No randomization method was used and data were not analyzed blindly. No statistical methods were used to predetermine sample size. The sample sizes in this study are in general similar to those employed in the field. The variance was similar between groups that were statistically compared. In all figures, *p* values are indicated as * < 0.05; ** < 0.01; *** < 0.001.

**Data availability.** The raw microarray data have been deposited in the Gene Expression Omnibus under ID codes GSE90700. Sequencing data for epigenetic signatures were retrieved from GSE55174 and GSE52464[14].

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

## Acknowledgements

We are grateful to Drs K. Kawakami, S. Kaneko and H. Nishihara for reagents; to Dr. N. Miyasaka for help with the zebrafish experiment; and to Dr. R. Mizuguchi and members of the RIKEN BSI Research Resource Center for the microarray experiment. We thank I. Rodriguez for helpful discussions and critical reading of the manuscript. T.I. was a Research Fellow of the JSPS. This work was supported in part by grant support from MEXT KAKENHI (Grant Numbers 20570208, 25660289, 16K07366 to J.H., 17K14932 to T.I.), from JSPS KAKENHI (Grant Number 21200010 to J.H.), from the ERATO Touhara Chemosensory Signal Project (JST ERATO Grant Numbers JPMJER1202 to Y.Y. and K.T.), and from the Senri Life Science Foundation, the Inamori Foundation, the Sumitomo Foundation, the Kurata Memorial Hitachi Science and Technology Foundation to J.H.

## Author contributions

T.I., C.K., D.S., T.E., Y.Y. and J.H. carried out all the wet-lab experiments. T.I., Y.N. and H.S. performed the bioinformatics analyses. T.I., K.T., Y.Y. and J.H. conceived the project and wrote the paper.
