## [Peer Review File · Nature Communications]

Reviewers' comments:

Reviewer #2 (Remarks to the Author):

The molecular mechanisms by which olfactory sensory neurons (OSNs) in the nose select to express a single olfactory receptor (OR) gene from up to 2,000 alternatives in the genome remain poorly understood. Some progress has been made over the past 5 years through the identification of cis-regulatory locus-specific (OR cluster-specific) enhancers and the description of epigenetic alterations that occur during OSN ontogeny. According to the prevailing model, singular OR expression is achieved by a hierarchical process that includes the nuclear aggregation of, and trans-interaction between, the various enhancer elements in a single OSN. However, only few of these enhancers have been studied in detail.

The work presented by Iwata and coworkers adds new and conceptually important aspects to the slowly emerging picture of OR gene choice. The authors report the identification and functional characterization of an additional cis-regulatory element, the J enhancer, which controls expression of the subset of class I OR genes that are contained within a single large cluster on mouse chromosome 7. Through bioinformatic, transgenic, and knock-out approaches the authors conclusively demonstrate that the J element has a regulatory impact on nearly the entire genomic range of the OR cluster and is strictly required for the expression of about half of the 158 class I genes. The study also provides clear evidence for the fact that the J element operates strictly in cis, does not extend activity to class II ORs on the same chromosome, and determines allelic exclusion among maternal and paternal alleles of class I ORs in a single OSN.

The current concept of OR gene choice suggested that during an early event a stochastic choice is made between several, if not all, enhancers to direct gene expression to a given OR gene cluster. However, this model does not account for other aspects of OR expression, such as zonal or class restriction, suggesting that additional regulatory mechanisms may be superimposed. Prior OR deletion experiments have demonstrated that class I and II OR genes are expressed by distinct OSN subtypes in the dorsal OE and that second OR gene choice is not restricted to genes controlled by the same enhancer element or to OR genes that reside in the same cluster. Thus, in principle, no coordination between enhancer elements would be required to achieve a distinction between class I and II ORs.

What is most interesting about the study presented in this ms is the finding that class I OR genes, similar to class II ORs, utilize the same enhancer-based mechanism to select OR gene expression at the cluster level. It is also interesting to see that expression from J enhancer transgenes is restricted to the dorsal OE where class I ORs are expressed. However, further studies will show whether this restriction is OSN subtype specific or a property of the enhancer. The chromosomal range of 3 MB over which the enhancer influences OR gene expression is just stunning and unprecedented for other enhancer systems as well.

The ms is clearly and thoughtfully written and presents the premises, experimental outcomes, conceptual framework, and implications in a way that makes the ms accessible to readers from outside the field. The findings are of general interest because the molecular process of OR gene choice includes conceptually novel mechanisms to coordinate expression of a large number of similar genes to generate cellular diversity and which are different from mechanisms in related systems, such as the immune system. For the olfactory community the data presented in the ms challenges existing concepts and offers new and fresh directions to develop revised experimental

models and hypotheses. Therefore, the work presented in this ms adds important and colourful strokes to the picture describing the largely enigmatic process of OR gene choice. I would like to congratulate the authors on a beautifully conducted, solid, and comprehensive piece of work that very well merits publication.

Minor points, which may need attention:

On page 14, last paragraph, the authors state that the Milos enhancer and the J element are just 0.9 kb apart. In as much would this close proximity make Milos an independent enhancer element?

In the same context, it did not become clear to me what the Δ J-Milos experiment contributes to the study and which sequences were actually deleted. The text reads as if only the genomic region between J and Milos would have been deleted, which is not a very telling experiment. In general, it is my personal opinion that it is not favorable to include "data not shown" experiments. In this respect, the term "unpublished data" that the authors use suggests that they will publish these very data and that they actually observed something in the Δ J-Milos line.

Figure 4 showed some strange character formatting in the pdf on several viewers, which makes part of the figure illegible

Reviewer #3 (Remarks to the Author):

Understanding "one receptor-one neuron" rule in the olfactory sensory neurons is fundamental to understand mammalian olfaction. Despite some high-impact publications, we still do not understand how cis regulatory elements regulate transcription of individual odorant receptors. Previously, other groups identified multiple locus control region-like enhancer elements (such as H and P), each of which is required for a few odorant receptors near the element.

Using an elegant combinations of transgenic mouse and sequence comparison approaches, this work identified a new cis-element (J element) that appears to specifically regulate expression of a part class I odorant receptors clustered at a single location of the chromosome. The experiments and analyses were generally conducted with adequate care and accuracy.

The novelty of this work is that the authors identified a cis regulatory sequences specific to Class I odorant receptors, whereas those elements previously identified regulate Class II odorant receptors. The Class I receptors are distinct from Class II in receptor in their evolutionary history as well as their chromosomal localization. The mechanism that J element dictate Class I receptor expression does not appear to be based on its chromosomal locations but its unique sequence including the "AAACTTTTC" motif in addition to the other motifs shared with previously identified enhancers such as H- P- and the "Greek Island" elements. However, the authors' analysis falls a bit short from demonstrating how this element ensures specificity. I have some recommendations to improve the quality of the work.

Major Points:

1. It is not clear to me how "AAACTTTTC" sequence motif acts specificity on Class I gene regulation. Epigenetic landscape of J element appears to be different from previously characterized elements is intriguing. But the authors did not describe how J element is marked. Meta analysis of available data (eg GSE55174) should provide useful information regarding epigenetic marks and open chromatin status of J element.

2. It sounds odd to me that a specific "cis" element regulates expression of a subset of receptors defined by "protein" sequences. In the Discussion, the authors appear to make the case using an example of a couple of Class I receptors located outside the cluster. A more careful discussion is

warranted here.

3. Though I agree with the authors that the majority of class I genes were affected by delta J, it is not clear how many receptors show abolished expression and how many genes were only moderately reduced in numbers of positive neurons. The authors should provide more detailed analysis to quantify the effects of delta J on class I gene expression as a whole.

4. It is an interesting observation that delta J effects on Class I receptor do not seem to depend on the vicinity of the receptors to the element. But the authors did not provide sufficient explanations.

5. Data using Delta J-Milos mice should be fully described.

6. The fate of the decreased ClassI expressing olfactory neurons in delta J/delta J was not investigated. Do the authors observe increased cell death?

7. Off target effects of the CRISPR gRNAs were not adequately discussed. The authors stated, "We backcrossed one founder to C57BL/6, and established ΔJ mice line #8 with a 1982 bp full deletion (mm10, chr7:102,509,690 - 102,511,671) of the J element.". How many generations of backcrosses the authors conducted to eliminate any potential off target mutations?

Minor Points:

1. Some of the fonts in a couple of figures, including Figs 4e, 4f and 6a, are distorted.

Response to the reviewers' comments

First of all, we would like to thank the reviewers' comments in shaping manuscript and improving its strength. Please note all modifications have been colored in blue in the revised manuscript.

Reviewer #2

Minor points

2-1) On page 14, last paragraph, the authors state that the Milos enhancer and the J element are just 0.9 kb apart. In as much would this close proximity make Milos an independent enhancer element? In the same context, it did not become clear to me what the Δ J-Milos experiment contributes to the study and which sequences were actually deleted. The text reads as if only the genomic region between J and Milos would have been deleted, which is not a very telling experiment. In general, it is my personal opinion that it is not favorable to include "data not shown" experiments. In this respect, the term "unpublished data" that the authors use suggests that they will publish these very data and that they actually observed something in the Δ J-Milos line.

[Redacted]

After careful consideration of the reviewer's comments, we would like to revise the manuscript to delete the Δ J-Milos experiment from the Discussion, because of following reasons;

- 1) Results of the Δ J-Milos experiment only suggest that an additional enhancer element (or elements) other than the Milos exists in the class I cluster.
- 2) We cannot not exclude the possibility that the Milos functions as an enhancer for class I genes (or class II genes). Alternatively, because the Milos is in close proximity to the J element, it is possible to speculate that the Milos may function in cooperation with the J element. In any case, functional analyses of the Milos such as transgenic reporter assays and knockout experiments are required to elucidate the function of the Milos.
- 3) Because we are currently working on identification and characterization of additional enhancer elements in the class I cluster, including the Δ Milos experiment, we would like to reserve the Δ J-Milos experiment for our future studies.

As pointed out by the reviewer, the Δ J-Milos experiment described in the Discussion does not contribute to this study greatly, rather it may dilute the main point of the study and confuse broad readers. Accordingly, we revised the manuscript to discuss possible additional enhancer elements, and we believe that this revision does not affect the importance and impact of this study.

- We revised the Discussion, **page 16, line 9 – 10.**

2-2) Figure 4 showed some strange character formatting in the pdf on several viewers, which makes part of the figure illegible

We corrected these errors and confirmed no strange character formatting. To avoid errors during the process of submission, we provide figure files separately.

Reviewer #3

Major points

3-1) It is not clear to me how “AAACTTTTC” sequence motif acts specificity on Class I gene regulation. Epigenetic landscape of J element appears to be different from previously characterized elements is intriguing. But the authors did not describe how J element is marked. Meta analysis of available data (eg GSE55174) should provide useful information regarding epigenetic marks and open chromatin status of J element.

To understand the molecular mechanisms underlying J-element dependent class I gene regulation, we are now screening transcription factors which bind to the “AAACTTTTC” sequence motif by yeast one-hybrid system as well as mutagenesis studies into the conserved motifs. We believe that our future studies will provide insights into this issue.

However, we noticed that our expression of “specific” in the Results (page 13) were inappropriate. Previous genetic experiments support the view that OSNs are fated to express a single OR gene from the repertoire of class I or class II genes. A given OSN cannot select indiscriminately between class I and class II genes, but is restricted by lineage or cell type to choose a class I or class II enhancer/promoter (Hirota *et al.*, Mol Cell Neurosci, 2007; Bozza *et al.*, Neuron, 2009). Therefore, it is conceivable that OSN-lineage/type determines the specific activation of class I or class II enhancers. We revised the manuscript as follows;

- We deleted “to investigate class I-specific enhancer activity of the J element” from **page 13, line 1.**
- We revised “this motif may be responsible for class I-specific gene expression” to “this motif may be responsible for the characteristic features of class I gene expression” in **page 13, line 23.**

According to the reviewer’s comment, we analyzed epigenetic landscape of the J element using sequence data retrieved from GSE55174 and GSE52464 (single data for DHS and H3K79me3; two replicates for H3K27ac, H3K4me1, and H3K27me3 in Gene Expression Omnibus) and compared with that of class II elements, the H, P, and Lipsi.

We revised the manuscript and provided these data as Supplementary Fig. 5.

- We added “Indeed, meta analysis of the sequencing data of GSE55174 and GSE52464 (Gene Expression Omnibus) used to identify 35 OR enhancer elements reveals that epigenetic landscape of the J element is different from previously characterized enhancer elements for class II genes (Supplementary Fig. 5)”, **page 15, line 19 – 22.**

- We added **Supplementary Fig. 5**, which displays epigenetic landscape of the J element along with class II enhancers, the H, P, and Lipsi.

3-2) It sounds odd to me that a specific “cis” element regulates expression of a subset of receptors defined by “protein” sequences. In the Discussion, the authors appear to make the case using an example of a couple of Class I receptors located outside the cluster. A more careful discussion is warranted here.

During evolution, class I genes have been retained in a single cluster on a single chromosome in all mammalian species, whereas class II genes spread over multiple clusters on multiple chromosome. Why the class I genes did not migrate outside of the cluster has remained a mystery in OR evolution. In the Discussion, we discussed our hypothetical model account for the single class I cluster. We also tested our hypothesis using two “class I” genes, which were exceptionally identified outside the class I cluster only in the mouse genome. Although the classification of OR class is based on amino acid sequences, their nucleotide sequences are highly identical to two class I genes residing inside the cluster. Thus, it is conceivable that these exceptional class I genes were duplicated from the class I cluster. We analyzed their expression in the wild-type MOE, and found that these exceptional class I genes are not or barely expressed, suggesting ongoing pseudogenization of these genes as well as supporting our hypothesis.

According to the reviewer’s comment, we carefully revised the Discussion.

- We revised the Discussion, from **page 16 line 14 to page 17 line 7**.

3-3) Though I agree with the authors that the majority of class I genes were affected by delta J, it is not clear how many receptors show abolished expression and how many genes were only moderately reduced in numbers of positive neurons. The authors should provide more detailed analysis to quantify the effects of delta J on class I gene expression as a whole.

Because it is not practical to analyze the effects of ΔJ on all the OR genes by ISH, we chose 18 OR genes as representatives (12 class I genes, 2 atypical class I genes, 4 class II genes). To comprehensively analyze the effects of ΔJ on all the OR genes, we directly compared the gene expression profiles of ΔJ and littermate wild-type mice using exon microarray analysis. All detailed quantitative data including the fold change (FC) value was provided in Fig. 4c and Supplementary Table 2. If FC criteria of ≤ -2.5 (severely reduced genes) and of > -2.5 (moderately reduced) are applied to the microarray data, 12 class I genes were severely reduced, and 63 genes were moderately reduced (Total 75 genes were significantly reduced; $FC < -1.3$, $p < 0.05$. None of the tested class I genes was abolished in ΔJ mice as shown in Supplementary Table 3). However, I am afraid that this criterion is subjective, and we would not like to add these in the revised manuscript.

Instead, we added the quantitative data of the ISH for the 18 OR probes as Supplementary Table 3 along with the microarray data of the FC values so that one can estimate the degree of the reduction class I gene expression from the microarray data (Supplementary Table 2). Because the FC value of microarray analysis was well correlated with the changes in the number of OSNs

analyzed by ISH analysis (Fig. 4f, $r = 0.87$), we believe that the FC values represent the levels of reduction in the number of OSNs.

- We added “Supplementary Table 2” to **page 9, line 23**.
- We added a new table “Supplementary Table 3” to **page 10, line 17**.

3-4) It is an interesting observation that delta J effects on Class I receptor do not seem to depend on the vicinity of the receptors to the element. But the authors did not provide sufficient explanations.

Although ΔJ showed a slightly graded effect on the expression of class I OR genes that was commensurate with genomic distance (log₂-fold change vs. genomic distance from the J element; Pearson correlation coefficient $r = 0.42$, $p = 9.7 \times 10^{-7}$) (Fig 4)), the J element regulates most class I genes expression by exerting an effect over ~3 megabase whole cluster, suggesting that the J element can access most class I genes through the cluster and transcription factors that bind to the J element bring it into proximity of target promoters by looping, which is thought to be mediated by cohesion and other protein complexes.

According to the reviewer’s comment, we revised the Discussion and provide possible explanations.

- We added “One possible explanation for an extraordinary long-range regulation of the J element is that class I genes throughout the cluster can be accessible by the J element, which is established by the organization of 3D chromatin structure of this locus. The probability of class I OR gene choice depends on 3D chromatin structure rather than linear genomic distance” to the Discussion, **page 12, line 12 – 15**.

3-5) Data using Delta J-Milos mice should be fully described.

We revised the Discussion as mentioned above. Please see our reply #2-1.

3-6) The fate of the decreased Class I expressing olfactory neurons in delta J/delta J was not investigated. Do the authors observe increased cell death?

According to the reviewer’s comment, we analyzed and quantified apoptotic cells in the MOE of ΔJ mice. We revised the manuscript accordingly.

- We added “A massive reduction of the expression levels of class I mRNAs in ΔJ mice indicates that the number of class I genes expressing OSNs is decreased. To investigate the fate of the decreased class I OSNs in ΔJ mice, we quantified apoptotic cells in the dorsal MOE using anti-active caspase-3 antibody. There was no significant difference in the number of apoptotic cells between ΔJ and wild-type mice (18.5 ± 4.31 , $n=3$ for wild-type; 19.2 ± 3.97 , $n=3$ for ΔJ ; mean \pm SEM, $p = 0.91$), suggesting that the deletion of the J element does not result in cell death. Because there was no significant effect of ΔJ on class

II genes expression, it is conceivable that OSNs that fail to express class I genes by ΔJ arrest neuronal differentiation rather express class II genes as secondary choice. Although the decreased number of class I OSNs might result in a thinner dorsal MOE, there was no morphological difference in the dorsal MOE between ΔJ and wild-type mice. Because OSNs expressing class I and class II genes are intermingled within the dorsal MOE, the reduction in the number of class I OSNs could not be detected morphologically” to the Results, **page 11, line 1 – 12.**

3-7) Off target effects of the CRISPR gRNAs were not adequately discussed. The authors stated, “We backcrossed one founder to C57BL/6, and established ΔJ mice line #8 with a 1982 bp full deletion (mm10, chr7:102,509,690 - 102,511,671) of the J element.” . How many generations of backcrosses the authors conducted to eliminate any potential off target mutations?

According to the reviewer’s comment, we provided detailed information regarding “off-target effects” and revised the Method. We also analyzed possible off-target sites by direct sequencing of genomic DNA of ΔJ founder mice, and confirmed that there were no indel mutations in these sites.

- We added detailed information to the Methods, **page 20 line 11 - 17.**
- We provided **Supplementary Table 5** (Off target analysis for ΔJ mice).

Minor Points

3-8) Some of the fonts in a couple of figures, including Figs 4e, 4f and 6a, are distorted.

We corrected these errors and confirmed no strange character formatting. To avoid errors during the process of submission, we provide figure files separately.

Figures and Tables

1. No modifications in the main Figures.

2. We added following materials;

Supplementary Figure 5 | An epigenetic signature for the J and class II enhancer elements.

Supplementary Table 3 | ISH analysis in ΔJ mice.

Supplementary Table 5 | Off-target analysis for ΔJ mice.

[Redacted]

REVIEWERS' COMMENTS:

Reviewer #2 (Remarks to the Author):

The authors have appropriately addressed my previous (minor) concerns in the revised version of the ms. I fully support the authors decision to exclude the 'preliminary' data on the ΔJ -Milos mutation as it does not add significantly to the main finding of the paper and because additional work that goes beyond the presented studies is needed to work out the details of the relationship between the J enhancer and additional regulatory elements within the cluster.

I can only reiterate my previous assessment of the work presented by Iwata and coworkers, which adds important new insight to the complex and largely enigmatic process of regulating OR gene choice at the genomic, chromosome and gene level. The study is very well executed, complete, insightful, and exciting beyond the olfactory field. The ms itself is dense but clearly written and presents an interesting new phenomenon that is supported by hard evidence. Again, I would like to congratulate the authors on such a beautiful and comprehensive piece of work that includes a multitude of different analysis to link their finding to related experimental designs that have been utilized in the past to shed light onto the process.

Reviewer #3 (Remarks to the Author):

The authors satisfactory addressed all of my previous comments.